# MEDUSA: Motion Elimination in Diffusion Using Spectral Attack

**Hongwei Yu** [* 1]   **Daoqing Zha** [* 1]   **Xinlong Ding** [1]   **Jiawei Li** [1]   **Junbao Zhuo** [1]   **Qiankun Liu** [1]   **Huimin Ma** [1]
**Jiansheng Chen** [✉ 1]

https://daoqingzha.github.io/medusa

## Abstract

With the widespread application of Video Diffusion Models (VDMs), video synthesis has achieved remarkable temporal dynamics. Image-to-Video (I2V) generation allows users to provide reference images, which enables attackers to inject adversarial noise into these conditions. Due to the robust spatio-temporal priors in VDMs, conventional frame-level attacks merely induce superficial artifacts and struggle to suppress the synthesis of motion semantics. In this work, we approach the problem by exploring the underlying mechanism of temporal dynamics. We reveal that the static video manifests as a temporal rank collapse, a degenerate state characterized by rank-1 degeneracy within the temporal attention matrix. Guided by this insight, we propose Motion Elimination in Diffusion Using Spectral Attack (MEDUSA) to freeze the video. It minimizes the nuclear norm of the attention matrix to induce the temporal rank collapse. This objective circumvents the vanishing gradient problem encountered when directly imposing a rigid temporal mapping on the attention matrix. Furthermore, we provide a mathematical analysis of this phenomenon and the gradient vanishing problem during the optimization. Experiments confirm that MEDUSA achieves excellent performance and validates the effectiveness of spectral constraints.

## 1. Introduction

The rapid proliferation of Video Diffusion Models (VDMs) (Xing et al., 2024a; Blattmann et al., 2023; Yang et al., 2025; HaCohen et al., 2024) represents a significant leap in generative artificial intelligence. Unlike static image generators, VDMs utilize temporal attention mechanisms (Bertasius et al., 2021) to model complex dependencies across frames, enabling temporally coherent dynamics. Image-to-Video (I2V) generation animates a static reference image into a coherent video sequence, which also raises safety concerns when protected images can be animated into misleading or unauthorized videos. Despite their impressive capabilities, the adversarial robustness of VDMs remains underexplored (Ma et al., 2026; Li et al., 2025b).

Most attack methods are designed for image generation (Salman et al., 2023; Shan et al., 2023; Yu et al., 2024). Naive transfers of image-based attacks to VDMs fail to arrest semantic motion, as they do not target the temporal priors of the model. Consequently, these attacks typically disrupt visual fidelity in generated videos but fail to produce a static effect. In the context of I2V, adversarial attacks typically inject perturbations into the reference image condition. Gui et al. (Gui et al., 2025) are the first to propose an attack targeting VDMs for I2V generation, but their primary objective is to degrade generation quality. This results in I2VGuard mainly producing ghosting-like artifacts and fails to restrict motion. Recent video-specific approaches, such as Vid-Freeze (Chowdhury et al., 2025), attempt to inhibit motion by minimizing the Frobenius norm of the temporal attention matrix. Nevertheless, this work merely imposes a penalty on the magnitude of element-wise weights, which functions as a form of energy suppression. It lacks an exploration of the underlying mechanisms of temporal dynamics.

In this work, we begin with a theoretical analysis. We posit that the richness of temporal dynamics in video generation is driven by the spectral properties of the temporal attention matrix. Specifically, the generation of rich motion requires a high-rank attention structure to model diverse temporal dependencies. Conversely, a static video corresponds to a $rank$-1 degeneracy in the attention matrix, which we refer to as **Temporal Rank Collapse** (Dong et al., 2021). We provide both statistical evidence and mathematical proofs for this observation. Statistical results indicate that when the model is induced to generate static videos, the attention matrix exhibits a distinct vertical stripe pattern, revealing that

---

*Equal contribution  [1]University of Science and Technology Beijing, Beijing, China. Correspondence to: Jiansheng Chen <jschen@ustb.edu.cn>.

*Proceedings of the 43rd International Conference on Machine Learning*, Seoul, South Korea. PMLR 306, 2026. Copyright 2026 by the author(s).

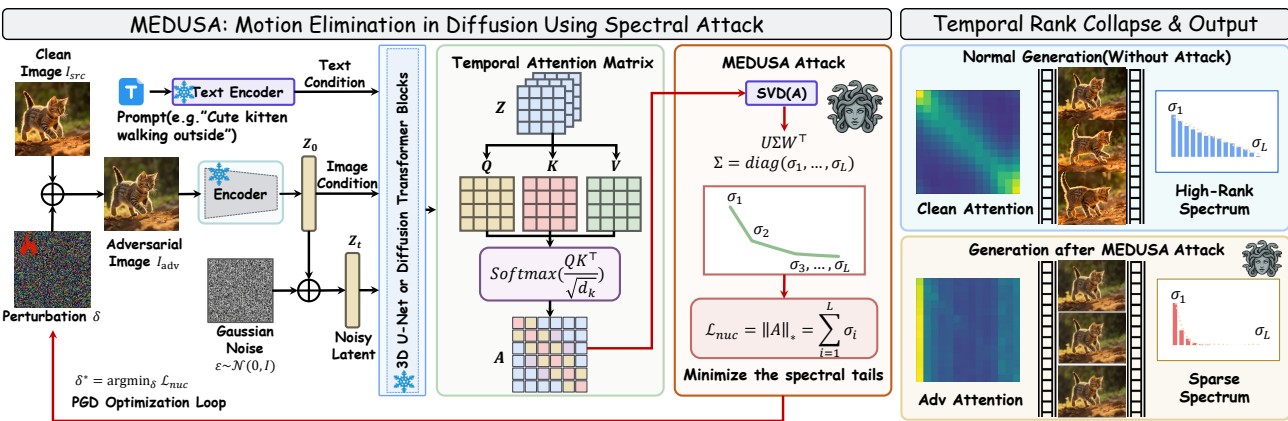

*Figure 1.* **Overview of the MEDUSA framework.** We optimize the adversarial perturbation by minimizing the Nuclear Norm of the temporal attention matrix. This objective suppresses the trailing singular values to induce temporal rank collapse. Consequently, the attention structure transforms into a Rank-1 vertical stripe pattern, which effectively eliminates temporal dynamics.

all temporal query frames are attending to the same specific frames. Mathematically, this implies that the row vectors of the attention matrix become linearly dependent, leading to a rank-1 degeneracy. Furthermore, we theoretically prove that the rank-1 collapse of the attention matrix is the cause of the loss of temporal dynamics. Crucially, we establish a spectral bound to guarantee that optimizing towards a low-rank state can effectively eliminate motion. We prove that the deviation from a static video is upper-bounded by the trailing singular values of the temporal attention matrix.

Guided by these findings, we define the optimization objective as reducing the rank of the temporal attention matrix. In practice, directly imposing a rigid mapping, such as forcing all frames to attend to the first frame, frequently results in attack failure. We identify this as a gradient dynamics bottleneck where hard constraints trigger the vanishing gradient problem, and we provide a mathematical analysis to substantiate this observation. To address this issue, we introduce **MEDUSA** (**M**otion **E**limination in **D**iffusion **U**sing **S**pectral **A**ttack), as illustrated in Figure 1. MEDUSA employs the nuclear norm, defined as the sum of the singular values (i.e., the spectrum) of the attention matrix. By minimizing this norm, MEDUSA directly suppresses the spectrum, thereby forcing the trailing singular values of these matrices towards zero (Wang et al., 2020). This process induces temporal rank collapse, thereby effectively reducing the temporal variation of the generative process (Li et al., 2026). Notably, such a spectral formulation circumvents the vanishing gradient problem encountered when directly imposing a rigid temporal mapping on the attention matrix.

We evaluate our method on various VDMs with both U-Net and DiT backbones. Extensive experiments are performed to verify the effectiveness of our attack. Compared with baselines, MEDUSA achieves superior motion elimination performance. The main contributions are as follows.

- We identify the underlying mechanism of temporal dynamics through a spectral lens. Specifically, we define **temporal rank collapse** as a rank-1 degeneracy within the temporal attention matrix.

- We propose **MEDUSA**, which utilizes the nuclear norm to induce temporal rank collapse via a spectral attack. This objective circumvents the gradient bottlenecks encountered by existing rigid mapping methods.

- Extensive experiments on various VDMs demonstrate that **MEDUSA** achieves state-of-the-art freezing performance. Our approach significantly outperforms other methods in suppressing motion semantics.

## 2. Related Work

### 2.1. Video Diffusion Models

The transition from static image synthesis to high-fidelity video generation represents a significant milestone in generative modeling (Yuan et al., 2026). Modern video diffusion models (VDMs) typically extend Latent Diffusion Models (LDMs) by integrating temporal modules into 3D U-Net (Ronneberger et al., 2015) or Diffusion Transformer (DiT) (Peebles & Xie, 2023) architectures, which form the two main backbone families considered in our evaluation. Stable Video Diffusion (SVD) (Blattmann et al., 2023) adapts a pre-trained image diffusion model by incorporating temporal convolution and attention layers to leverage strong spatial priors. DynamiCrafter (Xing et al., 2024a) employs a dual-stream architecture to inject image features, which ensures better semantic preservation. CogVideoX (Yang et al., 2025) introduces a 3D causal variational autoencoder and an expert transformer to align text and video modalities, which improves long-range consistency. LTX-Video (HaCohen et al., 2024) implements a transformer-based architecture

with a high-compression Video-VAE (Kingma & Welling, 2014) to achieve faster generation.

## 2.2. Adversarial Attacks on Image Generation

As generative models become ubiquitous, adversarial techniques (Li et al., 2023b) have been developed to safeguard content against unauthorized manipulation. Photo-Guard (Salman et al., 2023) introduces pixel-level perturbations that disrupt the latent representations of input images, which aims to hinder downstream manipulation such as inpainting. Glaze (Shan et al., 2023) focuses on style protection by optimizing perturbations that shift images in feature space toward alternative style representations. Building on these concepts, MFA (Yu et al., 2024) exploits the model's high sensitivity to input noise statistics. By modeling the step-wise vulnerability during the reverse process, it optimizes perturbations to induce significant mean fluctuations in the predicted noise, which leads to generation collapse. However, these methods focus exclusively on spatial features. When directly applied to video generation models, they fail to disrupt the strong temporal priors encoded in the model, which results in generated videos that retain semantic motion despite high-frequency visual artifacts.

## 2.3. Adversarial Attacks on Video Generation

Compared to the image domain, attacks targeting video diffusion models remain relatively underexplored (Li et al., 2023a; Tan et al., 2026; Ding et al., 2024). In the Text-to-Video (T2V) domain, T2VAttack (Li et al., 2025a) investigates prompt robustness by employing greedy word substitution and iterative token insertion strategies to disrupt the semantic fidelity and temporal dynamics of generated videos. In the I2V domain, I2VGuard (Gui et al., 2025) adopts a distribution divergence strategy to maximize the distance between adversarial and clean attention maps, forcing the model to generate chaotic motion. More closely related to our work is Vid-Freeze (Chowdhury et al., 2025), which inhibits motion by minimizing the Frobenius norm of attention maps. However, Vid-Freeze relies on magnitude suppression, which simply reduces interaction energy without addressing the underlying spectral structure of motion.

## 3. Methodology

In this section, we present the theoretical foundation of MEDUSA. We first introduce video latent diffusion models and the temporal attention mechanism, and then state the adversarial attack setup (Section 3.1). We then analyze temporal dynamics from a spectral perspective and propose the *temporal rank collapse*, which connects motion suppression to spectral degeneracy (Section 3.2). Subsequently, we justify the nuclear norm as the optimal objective function by analyzing the gradient limitations of alternative approaches

(Section 3.3). Finally, we describe the complete attack procedure (Section 3.4).

## 3.1. Preliminaries

**Video Diffusion Models** Video Diffusion Models (VDMs) extend image generation by learning a data distribution $p(x)$ over video sequences $x \in \mathbb{R}^{L \times H \times W \times C}$, where $L$ denotes the number of frames (Rombach et al., 2022). To reduce computational complexity, a pre-trained encoder $\mathcal{E}$ compresses the pixel-space video into a latent representation $z_0 = \mathcal{E}(x)$. The diffusion process is modeled as a Markov chain that gradually corrupts $z_0$ into Gaussian noise $z_T \sim \mathcal{N}(0, I)$ over $T$ steps (Ho et al., 2020; Song et al., 2021). The forward transition at timestep $t$ is given by:

$$z_t = \sqrt{\bar{\alpha}_t}\, z_0 + \sqrt{1 - \bar{\alpha}_t}\, \varepsilon, \quad \varepsilon \sim \mathcal{N}(0, I), \quad (1)$$

where $\bar{\alpha}_t$ represents the noise schedule. The generative process relies on a learnable denoising network $\epsilon_\theta(z_t, t, c)$, typically a 3D-UNet or Diffusion Transformer, which predicts the noise component given the current state $z_t$, timestep $t$, and conditioning context $c$ (e.g., text or a reference image).

**Temporal Attention Mechanism** The core component enabling motion synthesis in VDMs is the Temporal Attention module (Xing et al., 2024b). Unlike spatial layers that process each frame independently, temporal attention models long-range dependencies across the frame axis. Consider an intermediate feature map $z \in \mathbb{R}^{B \times C \times L \times H \times W}$. To compute temporal interactions, the tensor is reshaped to fold spatial dimensions into the batch axis: $z_{in} \in \mathbb{R}^{(B \cdot H \cdot W) \times L \times C}$. The input is projected into Query ($\mathbf{Q}$), Key ($\mathbf{K}$), and Value ($\mathbf{V}$) matrices via learnable weights $W^Q, W^K, W^V \in \mathbb{R}^{C \times d_k}$ (Vaswani et al., 2017). The temporal attention matrix $\hat{\mathbf{A}} \in \mathbb{R}^{(BHW) \times L \times L}$ is computed as:

$$\hat{\mathbf{A}} = \text{softmax}\left(\frac{\mathbf{Q}\mathbf{K}^\top}{\sqrt{d_k}}\right), \quad z_{out} = \hat{\mathbf{A}}\mathbf{V}. \quad (2)$$

For our analysis, we define $\mathbf{A} \in \mathbb{R}^{L \times L}$ as the spatial average of $\hat{\mathbf{A}}$ over all spatial locations. Here, the element $\mathbf{A}_{i,j}$ quantifies the aggregate attention weight that frame $i$ assigns to frame $j$. This matrix $\mathbf{A}$ serves as the pivotal structure encoding the video's temporal dynamics, making it the primary target for our spectral analysis.

**Adversarial Attack Setup** Adversarial attacks (Goodfellow et al., 2015) seek an imperceptible perturbation $\delta$ added to the conditioning input $c$ (e.g., a reference image) to steer the model output toward a target behavior. In our setting, the objective is motion suppression. We formulate the attack as the constrained optimization problem:

$$\min_\delta \ \mathcal{L}_{\text{adv}}\big(\epsilon_\theta(z_t, t, c + \delta)\big) \quad \text{s.t.} \quad \|\delta\|_\infty \leq \epsilon, \quad (3)$$

where $\mathcal{L}_{\text{adv}}$ measures motion magnitude. This problem is solved using Projected Gradient Descent (PGD) (Madry et al., 2018). At iteration $k$, the perturbation is updated by:

$$\delta_{k+1} = \Pi_{\mathcal{B}_\epsilon}\left(\delta_k - \gamma \cdot \text{sign}(\nabla_{\delta_k}\mathcal{L}_{\text{adv}})\right), \qquad (4)$$

where $\gamma$ is the step size and $\Pi_{\mathcal{B}_\epsilon}$ projects onto the $\ell_\infty$ ball $\mathcal{B}_\epsilon = \{\delta : \|\delta\|_\infty \le \epsilon\}$. Our work focuses on designing a theoretically grounded $\mathcal{L}_{\text{adv}}$ based on the spectral properties of temporal attention matrix.

## 3.2. Temporal Rank Collapse

**Empirical Observation**    We analyze the spectral behavior of the temporal attention matrix $\mathbf{A} \in \mathbb{R}^{L \times L}$ within the core layers of the VDM. In standard motion synthesis, $\mathbf{A}$ typically exhibits a high-rank band-diagonal structure, reflecting diverse temporal dependencies. However, when the model generates a static video, we observe a structural phase transition where the diagonal pattern dissolves into vertical stripes (Figure 4(b)). This pattern indicates that for a given key frame index $j$, the attention weight is nearly invariant across query frames, i.e., $\mathbf{A}_{i,j} \approx \mathbf{A}_{k,j}$ for all $i, k$. We term this spectral degeneracy as the temporal rank collapse, which means the row vectors of the attention matrix become linearly dependent. Our adversarial objective is to actively induce this collapse to suppress temporal dynamics.

**Theoretical Formulation**    Motivated by the observation of vertical striations, we posit that motion elimination is equivalent to forcing the temporal attention matrix to satisfy a rank-1 constraint. We first validate this hypothesis in the ideal scenario.

Let $\mathbf{V} \in \mathbb{R}^{L \times d}$ denote the value features and $\mathbf{Y} = \mathbf{A}\mathbf{V}$ denote the output features.

**Proposition 3.1** (Condition for Temporal Invariance)**.** *By definition, the attention matrix $\mathbf{A}$ is a row-stochastic matrix. It contains non-negative elements, and the sum of each row is equal to 1. If $\text{rank}(\mathbf{A}) = 1$, the video features are strictly stationary over time, implying $\mathbf{y}_i = \mathbf{y}_j$ for all frames $i, j$.*

*Proof.* Given that $\text{rank}(\mathbf{A}) = 1$, the matrix $\mathbf{A}$ can be decomposed as the outer product $\mathbf{u}\mathbf{v}^\top$. Since $\mathbf{A}\mathbf{1} = \mathbf{1}$, we obtain $\mathbf{u}(\mathbf{v}^\top\mathbf{1}) = \mathbf{1}$. Defining the scalar $\lambda = \mathbf{v}^\top\mathbf{1}$, we obtain $\lambda\mathbf{u} = \mathbf{1}$. This necessitates that $\mathbf{u}$ is proportional to the all-ones vector $\mathbf{1}$. Consequently, the attention matrix simplifies to the form $\mathbf{A} = \mathbf{1}\mathbf{w}^\top$ for some vector $\mathbf{w}$.

Substituting this structure into the output equation $\mathbf{Y} = \mathbf{A}\mathbf{V}$, we have:

$$\mathbf{Y} = (\mathbf{1}\mathbf{w}^\top)\mathbf{V} = \mathbf{1}(\mathbf{w}^\top\mathbf{V}). \qquad (5)$$

The product $\mathbf{w}^\top\mathbf{V}$ yields a single vector in $\mathbb{R}^{1 \times d}$, which we denote as $\mathbf{f}_{global}$. Thus, $\mathbf{Y} = \mathbf{1}\mathbf{f}_{global}$, implying that every

row $\mathbf{y}_i$ of the output is an exact replica of $\mathbf{f}_{global}$, thereby proving the temporal invariance. The complete derivation is provided in Appendix A.1. □

While Proposition 3.1 establishes the theoretical ideal, strictly enforcing a perfect rank-1 degeneracy is numerically intractable in practical adversarial optimization. Consequently, the optimization typically yields an approximate low-rank state. Crucially, to guarantee that approaching this state effectively drives the video towards stasis, we derive a spectral bound. The following lemma proves that the deviation from a static state is constrained by the trailing singular values of the attention matrix.

**Lemma 3.2** (Spectral Bound on Deviation from Stasis)**.** *The original output is $\mathbf{Y} = \mathbf{A}\mathbf{V}$. Its static version is formulated as $\mathbf{Y}_{\text{static}} = \mathbf{A}_{\text{static}}\mathbf{V}$, where all rows of $\mathbf{A}_{\text{static}}$ are identical. The deviation of the output $\mathbf{Y}$ from this static state $\mathbf{Y}_{\text{static}}$ can be defined as $\|\mathbf{Y} - \mathbf{Y}_{\text{static}}\|_F$. It is upper-bounded by the spectral tail:*

$$\|\mathbf{Y} - \mathbf{Y}_{\text{static}}\|_F \le (1 + \sqrt{L})\|\mathbf{V}\|_2\sqrt{\sum_{k=2}^{L}\sigma_k(\mathbf{A})^2}, \quad (6)$$

*where $L$ is the number of frames and $\|\mathbf{V}\|_2$ is the spectral norm of the value features. $\sigma_k(\mathbf{A})$ denotes the $k$-th largest singular value of the temporal attention matrix $\mathbf{A}$.*

*Proof.* Let $\mathbf{A}_1$ be the optimal rank-1 approximation of $\mathbf{A}$ given by the Eckart-Young-Mirsky theorem (Eckart & Young, 1936). By the triangle inequality, the deviation decomposes into two terms:

$$\|\mathbf{Y} - \mathbf{Y}_{\text{static}}\|_F \le \|\mathbf{A} - \mathbf{A}_1\|_F\|\mathbf{V}\|_2 + \|\mathbf{A}_1 - \mathbf{A}_{\text{static}}\|_F\|\mathbf{V}\|_2. \tag{7}$$

The first term is strictly controlled by the spectral tail $\|\mathbf{A} - \mathbf{A}_1\|_F = \sqrt{\sum_{k=2}^{L}\sigma_k(A)^2}$. For the second term, we leverage the row-stochastic constraint $\mathbf{A}\mathbf{1} = \mathbf{1}$ to establish that $\|\mathbf{A}_1 - \mathbf{A}_{\text{static}}\|_F \le \sqrt{L}\|\mathbf{A} - \mathbf{A}_1\|_F$.

Combining these two components directly yields Eq. 6. A complete derivation is provided in Appendix A.2. □

Lemma 3.2 serves as the theoretical cornerstone of our method. It reveals that the deviation from stasis is governed by the spectral tail ($\sigma_{2:L}$). This theoretical insight implies that minimizing the spectral tail is a sufficient condition to force the video towards a static state. It motivates our adoption of the nuclear norm, which serves as a convex surrogate to suppress these trailing singular values.

## 3.3. Optimization Analysis and Objective Formulation

Guided by the theoretical bounds established in Lemma 3.2, our primary goal is to induce a rank-1 degeneracy in the

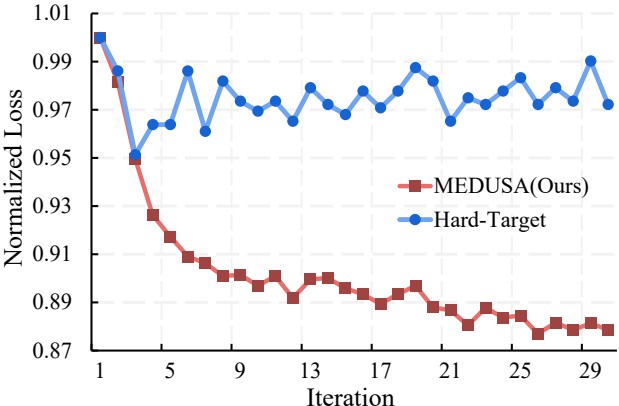

*Figure 2.* **Comparison of optimization convergence.** The Hard-Target baseline oscillates and does not converge reliably, whereas MEDUSA decreases smoothly and stably.

temporal attention matrix. A direct approach is to force the attention matrix to match a predefined static pattern. However, we observe that imposing such rigid constraints often leads to optimization failure. In this section, we identify the vanishing gradient problem within the Softmax mechanism as the root cause. Building on this, we propose nuclear norm minimization, which is the tightest convex surrogate for rank reduction (Recht et al., 2010).

**Gradient Vanishing in Hard Constraints**   Attempts to induce temporal rank collapse by compelling the attention matrix $\mathbf{A}$ to align with a static template confront an optimization problem. Specifically, consider Hard-Target attacks that force the attention matrix $\mathbf{A}$ to match a static template $\mathbf{A}^S$ (e.g., $\mathbf{A}^S_{i,1} = 1$) via mean squared error: $\mathcal{L} = \|\mathbf{A} - \mathbf{A}^S\|_F^2$. This formulation is numerically impeded by the softmax activation $\mathbf{A}_{i,j} = \exp(\mathbf{S}_{i,j} - \log \sum_k \exp(\mathbf{S}_{i,k}))$. By the chain rule, the gradient with respect to the pre-softmax scores $\mathbf{S}$ is:

$$\frac{\partial \mathcal{L}}{\partial \mathbf{S}_{i,j}} = \sum_k (\mathbf{A}_{i,k} - \mathbf{A}^S_{i,k}) \cdot \mathbf{A}_{i,k}(\mathbb{I}[k = j] - \mathbf{A}_{i,j}), \quad (8)$$

where $\mathbb{I}[k = j]$ is the Kronecker delta. The term $\mathbf{A}_{i,k}(\mathbb{I}[k = j] - \mathbf{A}_{i,j})$ is the Softmax Jacobian. As the optimization typically initializes from a highly dynamic state where $\mathbf{A}_{i,1} \approx 0$, this term approaches zero exponentially. This results in the vanishing gradient problem (Pascanu et al., 2013), causing the optimization to get stuck on local plateaus. This instability is illustrated in Figure 2.

**Spectral Relaxation via Nuclear Norm**   To resolve the aforementioned gradient bottlenecks, MEDUSA adopts a spectral approach (Tibshirani, 1996) by minimizing the nuclear norm of the temporal attention matrix. Formally, let $\mathbf{A} \in \mathbb{R}^{L \times L}$ denote the attention map extracted from the diffusion model. We perform Singular Value Decomposi-

tion (SVD) to factorize the matrix as $\mathbf{A} = \mathbf{U}\boldsymbol{\Sigma}\mathbf{W}^\top$, where $\boldsymbol{\Sigma} = \mathrm{diag}(\sigma_1, \ldots, \sigma_L)$ contains the singular values in descending order. Our optimization objective is defined as the sum of these values:

$$\mathcal{L}_{\mathrm{nuc}} = \|\mathbf{A}\|_* = \|\boldsymbol{\sigma}\|_1 = \sum_{k=1}^{L} \sigma_k. \quad (9)$$

Under the softmax constraint, the largest singular value remains unattenuated. Minimizing the nuclear norm aligns with the objective of reducing the spectral tail as formulated in Lemma 3.2. From an optimization perspective, the fundamental goal of reducing $\mathrm{rank}(\mathbf{A})$ is to minimize the $\ell_0$-norm of the singular value vector. However, the $\ell_0$-norm is non-convex. The nuclear norm serves as the tightest convex envelope of the rank function on the unit ball (Fazel, 2002; Candes & Recht, 2012), offering a tractable surrogate for minimization:

$$\mathrm{rank}(\mathbf{A}) = \|\boldsymbol{\sigma}\|_0 \xrightarrow{\text{Convex Relaxation}} \|\mathbf{A}\|_* = \|\boldsymbol{\sigma}\|_1. \quad (10)$$

Critically, this formulation induces spectral sparsity via a constant gradient pressure. The subgradient of the nuclear norm with respect to a singular value is governed by the sign function ($\partial \mathcal{L}/\partial \sigma_k \in \mathrm{sign}(\sigma_k)$), which maintains a constant magnitude even as $\sigma_k \to 0$. This drives the trailing singular values to zero, effectively suppressing temporal dynamics.

### 3.4. Algorithm

We formulate MEDUSA as an optimization problem over the conditioning input space. Given a clean reference image $x_{ref}$ and a pre-trained Video LDM $\epsilon_\theta$, our goal is to find an adversarial perturbation $\delta$ constrained by $\|\delta\|_\infty \leq \epsilon$ such that the generated video becomes static. The optimization process follows the Projected Gradient Descent (PGD) framework. To ensure attack efficiency, we target selected timesteps $t^*$ and temporal attention layers $l^*$, where motion semantics are most densely encoded. The complete procedure is detailed in Algorithm 1.

## 4. Experiments

### 4.1. Experimental Setup

**Datasets**   We construct a diverse benchmark dataset comprising 300 images for evaluation. The dataset is balanced across four semantic categories (Humans, Animals, Landscapes, and Objects) and includes both real-world photographs and AI-generated images. All images are resized to match the input resolution required by each target model.

**Models and Baselines**   To assess cross-architecture transferability, we employ Stable Video Diffusion (SVD) (Blattmann et al., 2023) and DynamiCrafter (Xing

**Algorithm 1** MEDUSA: Motion Elimination in Diffusion Using Spectral Attack

---

**Require:** Video LDM $\epsilon_\theta$, Reference Image $x_{ref}$.
**Require:** Parameters: Timestep $t^*$, Layer Index $l^*$, Budget $\epsilon$, Step Size $\gamma$, Iterations $K$, Token count $N = BHW \cdot$ Heads.
1: **Initialize:** $\delta \leftarrow \mathcal{U}(-\epsilon, \epsilon)$ {Random initialization}
2: $x_{adv} \leftarrow x_{ref} + \delta$
3: **for** $k = 1$ **to** $K$ **do**
4:     *// Latent Diffusion Preparation*
5:     $\mathbf{z}_0 \leftarrow \mathcal{E}(x_{adv})$ {Encode perturbed image}
6:     Sample noise $\varepsilon \sim \mathcal{N}(0, \mathbf{I})$
7:     $\mathbf{z}_{t^*} \leftarrow \sqrt{\bar{\alpha}_{t^*}}\mathbf{z}_0 + \sqrt{1 - \bar{\alpha}_{t^*}}\,\varepsilon$ {Diffuse to target step}
8:     *// Forward Pass & Spectral Extraction*
9:     $\mathbf{A} \leftarrow \mathcal{L}_{\text{attn}}(\epsilon_\theta(\mathbf{z}_{t^*}, t^*, x_{adv}); l^*)$ {Get matrix at $l^*$}
10:    $\mathbf{A} \leftarrow \text{Reshape}(\mathbf{A}) \in \mathbb{R}^{N \times L \times L}$
11:    *// Spectral Objective*
12:    $\mathcal{L}_{\text{nuc}} \leftarrow \frac{1}{N}\sum_{n=1}^{N}\|\mathbf{A}^n\|_*$ {$\mathbf{A}^n \in \mathbb{R}^{L \times L}$}
13:    *// PGD Update*
14:    $g \leftarrow \nabla_\delta \mathcal{L}_{nuc}$
15:    $\delta \leftarrow \delta - \gamma \cdot \text{sign}(g)$
16:    $\delta \leftarrow \Pi_{\mathcal{B}_\epsilon}(\delta)$ {$\mathcal{B}_\epsilon = \{\delta : \|\delta\|_\infty \le \epsilon\}$}
17:    $x_{adv} \leftarrow \text{Clip}(x_{ref} + \delta, 0, 1)$
18: **end for**
**Ensure:** Adversarial Image $x_{adv}$

---

et al., 2024a) as U-Net based models. We also include LTX-Video (HaCohen et al., 2024), which adopts a Diffusion Transformer (DiT) backbone, to verify that our method generalizes across both architectures. We compare our method against four baselines: PhotoGuard (Salman et al., 2023), MFA (Yu et al., 2024), I2VGuard (Gui et al., 2025), and Vid-Freeze (Chowdhury et al., 2025).

**Evaluation Metrics** We employ a comprehensive set of metrics to quantify both motion suppression and visual quality. To measure motion suppression, we report the Average Optical Flow magnitude (Farnebäck, 2003) and Dynamic Degree (Huang et al., 2024), which measures the percentage of pixels exhibiting significant motion. We also report Temporal SSIM to quantify temporal consistency across frames (higher is better). To assess visual quality and identity preservation, we report Subject Consistency (Caron et al., 2021), alongside standard perceptual metrics including FID (Heusel et al., 2017), SSIM (Wang et al., 2004), and LPIPS (Zhang et al., 2018).

**Implementation Details** Attacks are implemented using PGD with an $\ell_\infty$ constraint. Following the settings of (Chowdhury et al., 2025; Gui et al., 2025), we set the perturbation budget $\epsilon = 16/255$, step size $\gamma = 0.01$, and iterations $K = 50$. All experiments are conducted on 2 NVIDIA A800 (80GB) GPUs. Details are provided in Appendix B.

## 4.2. Main Results

**Quantitative Analysis** Table 1 presents a comprehensive comparison of different attack methods on SVD, DynamiCrafter, and LTX-Video. MEDUSA achieves state-of-the-art motion suppression across all three architectures. In particular, both Optical Flow and Dynamic Degree are substantially reduced compared to the clean baseline, consistently outperforming the competing methods. MEDUSA also improves Temporal SSIM, reaching an impressive 97.33% on LTX-Video. Across architectures, Subject Consistency remains competitive, indicating that MEDUSA suppresses temporal dynamics without compromising identity fidelity. Overall, these results demonstrate that MEDUSA effectively disrupts video generation by arresting temporal dynamics and producing near-static outputs.

**Qualitative Analysis** Figure 3 presents a qualitative comparison on a dynamic scene (a dog running in the rain). Existing image-based attacks (PhotoGuard, MFA) do not consistently suppress temporal dynamics: noticeable motion remains in both the foreground dog and the background vehicle. Compared to the clean baseline, these methods mainly introduce minor color shifts (MFA) or texture degradation (PhotoGuard). Since I2VGuard is designed for quality degradation, it indeed introduces observable structural distortions in specific frames. However, it remains ineffective in suppressing temporal dynamics. While Vid-Freeze effectively mitigates background dynamics, it induces anomalous structural distortions on the foreground subject, ultimately failing to achieve a truly static state. In contrast, MEDUSA achieves comprehensive suppression of temporal dynamics across both the foreground and background, yielding a perceptually static video sequence.

## 4.3. Spectral Analysis of Temporal Attention Matrix

To verify that the freezing effect is driven by our theoretical temporal rank collapse, we evaluate the spectral properties of the temporal attention matrix on SVD and DynamiCrafter before and after the attack. We employ two metrics to measure spectral sparsity. Band Energy Ratio (BER) (Child et al., 2019) quantifies the concentration of spectral energy within the principal components. A lower BER indicates a lower-rank structure. Effective Rank (ER) (Roy & Vetterli, 2007) is computed as the exponential of the spectral entropy and measures the effective dimensionality of the matrix. A value close to 1 indicates a rank-1 degenerate state. As shown in Table 2, both BER and ER decrease significantly after the attack. This quantitative evidence confirms that MEDUSA successfully guides the model into a state of temporal rank collapse. Crucially, this spectral degeneracy is correlated with the elimination of temporal dynamics, validating our hypothesis. We also provide a visualization of the SVD temporal attention matrix in Figure 4. Figure 4(a)

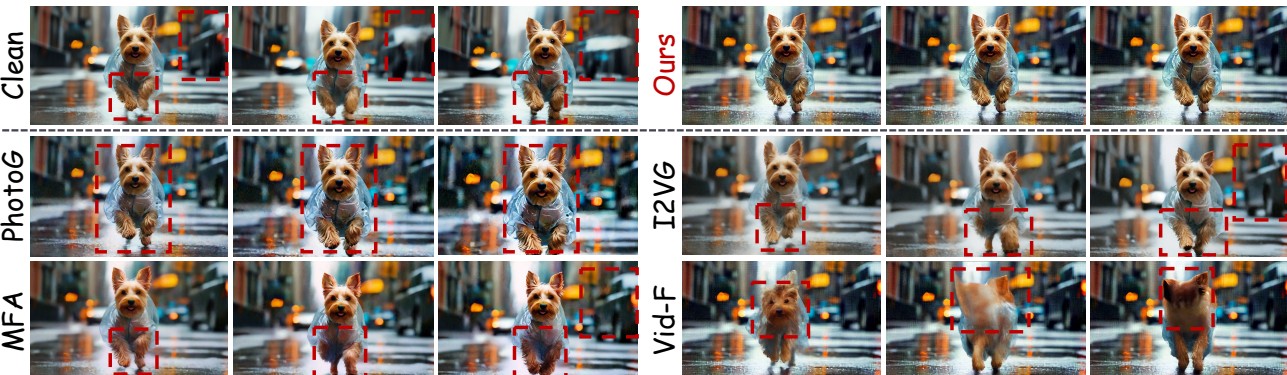

*Figure 3.* **Qualitative comparison of motion suppression.** While baselines fail to arrest motion (PhotoGuard, MFA) or introduce severe visual artifacts (Vid-Freeze, I2VGuard), MEDUSA achieves a comprehensive freeze. Note that our method preserves the semantic integrity of the subject (the dog) and background while effectively eliminating temporal variations.

*Table 1.* **Quantitative comparison on three diverse architectures.** We compare MEDUSA with baselines on SVD, DynamiCrafter, and LTX-Video. Lower is better for Optical Flow and Dynamic Degree, and higher is better for Temporal SSIM and Subject Consistency. All values are averaged over the evaluation set. Best results are highlighted.

| Model | Metric | Clean | PhotoGuard | MFA | I2VGuard | Vid-Freeze | Ours |
|---|---|---|---|---|---|---|---|
| SVD | Optical Flow (↓) | 2.7799 | 2.4745 | 2.3841 | 2.7248 | 1.9099 | **1.2246** |
| | Dynamic Degree (↓) | 43.20% | 37.64% | 34.62% | 40.19% | 27.36% | **13.72%** |
| | Temporal SSIM (↑) | 75.89% | 74.99% | 77.41% | 77.83% | 82.89% | **91.47%** |
| | Subject Consistency (↑) | **95.64%** | 94.10% | 94.26% | 94.15% | 95.10% | 95.27% |
| DynamiCrafter | Optical Flow (↓) | 2.1703 | 4.0358 | 1.7923 | 4.5556 | 1.0895 | **0.8085** |
| | Dynamic Degree (↓) | 32.57% | 57.21% | 28.79% | 54.69% | 21.20% | **19.82%** |
| | Temporal SSIM (↑) | 77.72% | 53.07% | 64.03% | 59.61% | 86.37% | **88.52%** |
| | Subject Consistency (↑) | 89.90% | 75.87% | 80.61% | 72.88% | 87.72% | **91.32%** |
| LTX-Video | Optical Flow (↓) | 0.7886 | 0.5132 | 0.5851 | 0.6768 | 0.4187 | **0.2867** |
| | Dynamic Degree (↓) | 23.84% | 15.47% | 17.81% | 20.98% | 12.13% | **7.98%** |
| | Temporal SSIM (↑) | 90.16% | 95.03% | 94.32% | 93.52% | 96.10% | **97.33%** |
| | Subject Consistency (↑) | 79.27% | **89.52%** | 88.37% | 85.53% | 85.67% | 86.93% |

displays the clean attention matrix. Figure 4(b) shows the matrix under static tuning, where the model's dynamic guidance parameter (controlling motion strength) is set to 0 (on a scale of 0–255). Figure 4(c) presents the matrix under MEDUSA. Figure 4(d) plots the singular value distribution. The attacked matrix shifts from a diagonal-dominant pattern to a structure characterized by vertical striations, suggesting that all frames uniformly attend to the same specific frames. After attacking, the spectral tails decay towards zero. These observations confirm that MEDUSA effectively induces temporal rank collapse, which serves as the spectral mechanism for motion elimination.

### 4.4. Robustness Analysis

We also evaluate our attack against three standard input purification defenses (Guo et al., 2018; Dziugaite et al.,

*Table 2.* **Spectral statistics of temporal attention.** We report BER and ER before and after the attack. For ER, we report the value along with the number of frames $L$ as ER / $L$.

| Model | Condition | BER(↓) | ER(↓) |
|---|---|---|---|
| SVD | Clean | 0.7471 | 7.74 / 14 |
| | Attack | **0.3862** | **2.96 / 14** |
| DynamiCrafter | Clean | 0.5569 | 6.04 / 16 |
| | Attack | **0.3495** | **3.40 / 16** |

2016): JPEG compression, Gaussian noise, and Bit-depth reduction. As summarized in Table 3, while these defense mechanisms induce a slight recovery in motion metrics compared to the no defense scenario, the generated videos remain predominantly static. This persistence suggests that the adversarial signal injected by MEDUSA is not merely

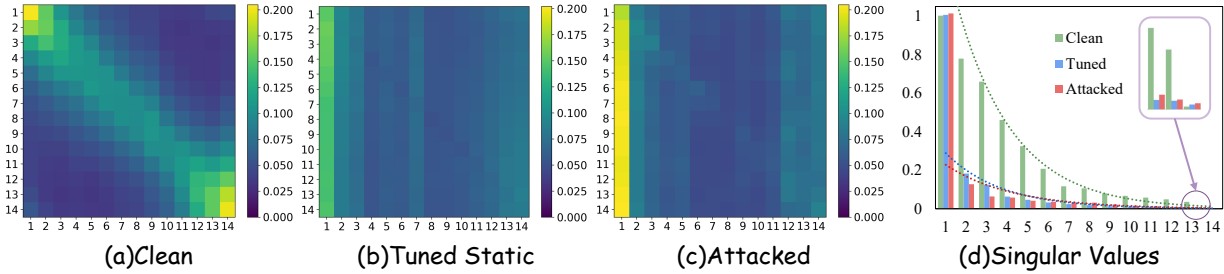

(a)Clean     (b)Tuned Static     (c)Attacked     (d)Singular Values

*Figure 4.* **Visualization of temporal attention matrices and singular values.** (a) The attention matrix of the clean video. (b) The matrix under static tuning. (c) The matrix after the MEDUSA attack. (d) The comparison of singular value distributions. MEDUSA induces a more columnar attention pattern and a faster decay in the singular values, consistent with a lower-rank regime.

*Table 3.* **Robustness against input transformations.** Attack performance under JPEG compression, Gaussian noise, and Bit-depth reduction. Even under these defenses, MEDUSA maintains significantly lower motion metrics compared to the Clean baseline.

| Defense Method | Optical Flow (↓) | Dynamic Degree (↓) |
|---|---|---|
| Clean Baseline | 2.7799 | 43.20% |
| MEDUSA (No Defense) | **1.2246** | **13.72%** |
| JPEG Compression | 1.7984 | 21.93% |
| Gaussian Noise | 1.7644 | 20.72% |
| Bit-depth Reduction | 1.4172 | 16.31% |

*Table 4.* **Ablation study on target temporal blocks.** Motion suppression across different attention blocks. Targeting the intermediate block (Block 5) yields the strongest motion suppression.

| Metric | Block 2 | Block 4 | Block 5 | Block 7 | Block 8 |
|---|---|---|---|---|---|
| Optical Flow (↓) | 2.7708 | 1.9095 | **0.5980** | 1.3216 | 1.1330 |
| Dynamic Degree (↓) | 55.04% | 21.13% | **7.23%** | 14.61% | 15.42% |

*Table 5.* **Comparison of optimization strategies.** Performance comparison between Hard-Target optimization and the spectral attack strategy. (OF: Optical Flow, DD: Dynamic Degree, T-SSIM: Temporal SSIM, SC: Subject Consistency).

| Method | OF (↓) | DD (↓) | T-SSIM (↑) | SC (↑) |
|---|---|---|---|---|
| Clean | 2.8010 | 46.78% | 74.01% | 95.08% |
| Hard-Target | 2.3325 | 36.91% | 76.99% | 93.88% |
| MEDUSA(Ours) | **0.5980** | **7.23%** | **94.81%** | **97.21%** |

*Table 6.* **Quantitative assessment of imperceptibility.** Comparison of visual quality metrics between MEDUSA and a random noise baseline.

| Method | FID (↓) | SSIM (↑) | LPIPS (↓) |
|---|---|---|---|
| Random noise | 12.6230 | 0.5454 | 0.5110 |
| MEDUSA (Ours) | **10.7146** | **0.8202** | **0.3162** |

superficial noise. Instead, minimizing the nuclear norm encourages a low-rank constraint that is relatively stable under these input transformations.

### 4.5. Ablation

**Ablation of Attack Blocks** To maintain computational feasibility under limited GPU memory, MEDUSA is designed to target specific temporal attention blocks rather than the entire network. To identify the optimal configuration, we conduct an evaluation on a subset of 50 randomly sampled reference images using SVD. We calculated the average attack efficacy across different blocks. Table 4 indicates that the intermediate-to-deep layers yield the most significant motion suppression. Consequently, we select these intermediate layers as our default target, achieving the optimal trade-off between performance and cost.

**Ablation of Optimization Choice** To validate the necessity of the nuclear norm, we compare MEDUSA against a Hard-Target baseline (i.e., directly minimizing the MSE loss to encourage the attention matrix to match a static iden-

tity matrix). As shown in Table 5, the Hard-Target method yields performance metrics comparable to the Clean baseline, demonstrating negligible motion suppression. This disparity corroborates our theoretical analysis. The Hard-Target method suffers from vanishing gradients.

**Imperceptibility of Our Attack** We further evaluate the imperceptibility of our adversarial examples. To assess perceptual fidelity, we compare MEDUSA against a baseline of random gaussian noise injected at half the budget (8/255) of our method. As shown in Table 6, we compute standard image quality metrics between the perturbed samples and their corresponding clean images. Quantitative results indicate that MEDUSA yields superior fidelity scores compared to random noise, demonstrating that our perturbations are more imperceptible. Additional visualizations are provided in the Appendix D.

## 5. Conclusion

In this work, we present **MEDUSA**, a theoretically grounded framework designed to attack Video Diffusion

Models in I2V task. We identify **temporal rank collapse**, characterized as a rank-1 degeneracy within the temporal attention matrix, as the condition for video stasis. Guided by this insight, MEDUSA minimizes the nuclear norm to suppress the trailing singular values of the attention spectrum. This optimization strategy effectively induces rank collapse while circumventing the gradient vanishing bottlenecks inherent in rigid hard-target constraints. Extensive experiments across various architectures confirm that MEDUSA achieves state-of-the-art motion elimination performance.

## Acknowledgment

This work was supported by the National Natural Science Foundation of China (62376024), the National Science and Technology Major Project (2022ZD0117902), the Beijing Natural Science Foundation (L257003) and the Fundamental Research Funds for the Central Universities (FRFTP-22-043A1).

## Impact Statement

This paper is intended to advance the field of computer vision. It involves no unethical considerations or adverse impacts, and is used solely for research purposes.

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

# A. Detailed Theoretical Proofs

## A.1. Proof of Proposition 3.1: Condition for Temporal Invariance

*Proof.* **Rank-1 Decomposition:** Since $\text{rank}(\mathbf{A}) = 1$, the matrix $\mathbf{A}$ can be expressed as the outer product of two non-zero vectors, $\mathbf{u} \in \mathbb{R}^L$ and $\mathbf{v} \in \mathbb{R}^L$:

$$\mathbf{A} = \mathbf{u}\mathbf{v}^\top \tag{11}$$

The element-wise definition is $A_{i,j} = u_i v_j$.

**Implication of Row-Stochasticity:** The Softmax operation ensures that each row of $\mathbf{A}$ sums to 1. Mathematically, this is expressed as:

$$\mathbf{A}\mathbf{1} = \mathbf{1} \tag{12}$$

Substituting the rank-1 decomposition into this constraint yields:

$$(\mathbf{u}\mathbf{v}^\top)\mathbf{1} = \mathbf{u}(\mathbf{v}^\top\mathbf{1}) = \mathbf{1} \tag{13}$$

Let $\lambda = \mathbf{v}^\top\mathbf{1} = \sum_{k=1}^{L} v_k$. The term $\lambda$ is a scalar constant. The equation becomes:

$$\lambda\mathbf{u} = \mathbf{1} \tag{14}$$

This equality implies that the vector $\mathbf{u}$ must be proportional to the all-ones vector $\mathbf{1}$. Specifically, every element of $\mathbf{u}$ must be equal to $1/\lambda$. Note that $\lambda$ cannot be zero, as that would imply $\mathbf{0} = \mathbf{1}$, a contradiction.

**Normalization and Structure of $\mathbf{A}$:** We substitute the derived expression for $\mathbf{u}$ back into the decomposition of $\mathbf{A}$:

$$\mathbf{A} = (\lambda^{-1}\mathbf{1})\mathbf{v}^\top = \mathbf{1}(\lambda^{-1}\mathbf{v})^\top \tag{15}$$

We now define the normalized probability vector $\mathbf{w} = \lambda^{-1}\mathbf{v}$. Consequently, the attention matrix simplifies to the specific form:

$$\mathbf{A} = \mathbf{1}\mathbf{w}^\top = \begin{bmatrix} 1 \\ 1 \\ \vdots \\ 1 \end{bmatrix} \begin{bmatrix} w_1 & w_2 & \dots & w_L \end{bmatrix} = \begin{bmatrix} w_1 & w_2 & \dots & w_L \\ w_1 & w_2 & \dots & w_L \\ \vdots & \vdots & \ddots & \vdots \\ w_1 & w_2 & \dots & w_L \end{bmatrix} \tag{16}$$

This structure confirms the "vertical stripe pattern" observed empirically in static videos. Every row of the matrix is identical to the probability distribution vector $\mathbf{w}^\top$. This means that regardless of the query frame index $i$, the attention mechanism assigns the exact same weight $w_j$ to key frame $j$.

**Output Invariance:** We now compute the output features $\mathbf{Y} = \mathbf{A}\mathbf{V}$:

$$\mathbf{Y} = (\mathbf{1}\mathbf{w}^\top)\mathbf{V} = \mathbf{1}(\mathbf{w}^\top\mathbf{V}) \tag{17}$$

The term $(\mathbf{w}^\top\mathbf{V})$ represents a vector-matrix multiplication resulting in a single row vector, which we denote as $\mathbf{f}_{\text{global}} \in \mathbb{R}^{1 \times d}$. Thus:

$$\mathbf{Y} = \begin{bmatrix} \mathbf{f}_{\text{global}} \\ \mathbf{f}_{\text{global}} \\ \vdots \\ \mathbf{f}_{\text{global}} \end{bmatrix} \tag{18}$$

This implies that every row $\mathbf{y}_i$ of the output is an exact replica of $\mathbf{f}_{\text{global}}$, thereby proving the temporal invariance. $\square$

## A.2. Proof of Lemma 3.2: Spectral Bound on Deviation from Stasis

*Proof.* Our goal is to bound the distance between the actual output $\mathbf{Y}$ and its static state $\mathbf{Y}_{\text{static}}$. We achieve this by using the optimal rank-1 approximation $\mathbf{A}_1$ as a bridge.

**The Optimal Rank-1 Bridge:** Let the Singular Value Decomposition (SVD) of $\mathbf{A}$ be $\mathbf{U}\boldsymbol{\Sigma}\mathbf{W}^\top$, with singular values $\sigma_1 \geq \sigma_2 \geq \dots \geq \sigma_L$. According to the Eckart-Young-Mirsky theorem, the optimal rank-1 approximation of $\mathbf{A}$ in the

Frobenius norm is given by the truncated SVD:

$$\mathbf{A}_1 = \sigma_1 \mathbf{u}_1 \mathbf{v}_1^\top, \quad \|\mathbf{A} - \mathbf{A}_1\|_F = \sqrt{\sum_{k=2}^{L} \sigma_k(\mathbf{A})^2}. \tag{19}$$

Using the triangle inequality, we bound the total deviation:

$$
\begin{aligned}
\|\mathbf{Y} - \mathbf{Y}_{\text{static}}\|_F &= \|(\mathbf{A} - \mathbf{A}_{\text{static}})\mathbf{V}\|_F \\
&\leq \|(\mathbf{A} - \mathbf{A}_1)\mathbf{V}\|_F + \|(\mathbf{A}_1 - \mathbf{A}_{\text{static}})\mathbf{V}\|_F \\
&\leq \|\mathbf{A} - \mathbf{A}_1\|_F \|\mathbf{V}\|_2 + \|\mathbf{A}_1 - \mathbf{A}_{\text{static}}\|_F \|\mathbf{V}\|_2.
\end{aligned}
\tag{20}
$$

The first term is directly given by Eq. 19. We now focus on bounding the second term $\|\mathbf{A}_1 - \mathbf{A}_{\text{static}}\|_F$.

**Bounding the Stochastic Adjustment Term:** Since $\mathbf{A}$ is element-wise non-negative, by the Perron-Frobenius theorem, the first right singular vector $\mathbf{v}_1$ can be chosen to be non-negative. Hence $\mathbf{v}_1^\top \mathbf{1} = \|\mathbf{v}_1\|_1 > 0$ and moreover $\mathbf{v}_1^\top \mathbf{1} \geq \|\mathbf{v}_1\|_2 = 1$. We define $\mathbf{A}_{\text{static}}$ by normalizing $\mathbf{v}_1$:

$$\mathbf{q} = \frac{\mathbf{v}_1}{\mathbf{v}_1^\top \mathbf{1}}, \quad \mathbf{A}_{\text{static}} = \mathbf{1}\mathbf{q}^\top. \tag{21}$$

By construction, $\mathbf{A}_{\text{static}}\mathbf{1} = \mathbf{1}$, ensuring it is row-stochastic.

Substituting the definitions of $\mathbf{A}_1$ and $\mathbf{A}_{\text{static}}$:

$$\mathbf{A}_1 - \mathbf{A}_{\text{static}} = \sigma_1 \mathbf{u}_1 \mathbf{v}_1^\top - \mathbf{1}\frac{\mathbf{v}_1^\top}{\mathbf{v}_1^\top \mathbf{1}} = \left(\sigma_1 \mathbf{u}_1 - \frac{\mathbf{1}}{\mathbf{v}_1^\top \mathbf{1}}\right)\mathbf{v}_1^\top. \tag{22}$$

Since $\|\mathbf{x}\mathbf{y}^\top\|_F = \|\mathbf{x}\|_2 \|\mathbf{y}\|_2$ and $\|\mathbf{v}_1\|_2 = 1$, we have:

$$\|\mathbf{A}_1 - \mathbf{A}_{\text{static}}\|_F = \left\|\sigma_1 \mathbf{u}_1 - \frac{\mathbf{1}}{\mathbf{v}_1^\top \mathbf{1}}\right\|_2. \tag{23}$$

Factoring out the scalar term:

$$\|\mathbf{A}_1 - \mathbf{A}_{\text{static}}\|_F = \frac{1}{\mathbf{v}_1^\top \mathbf{1}}\|\sigma_1 \mathbf{u}_1(\mathbf{v}_1^\top \mathbf{1}) - \mathbf{1}\|_2 = \frac{1}{\mathbf{v}_1^\top \mathbf{1}}\|\mathbf{A}_1 \mathbf{1} - \mathbf{1}\|_2. \tag{24}$$

We leverage the row-stochastic property of the original matrix $\mathbf{A}$ (i.e., $\mathbf{A}\mathbf{1} = \mathbf{1}$):

$$\|\mathbf{A}_1 \mathbf{1} - \mathbf{1}\|_2 = \|\mathbf{A}_1 \mathbf{1} - \mathbf{A}\mathbf{1}\|_2 = \|(\mathbf{A}_1 - \mathbf{A})\mathbf{1}\|_2. \tag{25}$$

Applying the consistent matrix norm property:

$$\|(\mathbf{A}_1 - \mathbf{A})\mathbf{1}\|_2 \leq \|\mathbf{A}_1 - \mathbf{A}\|_F \|\mathbf{1}\|_2 = \sqrt{L}\|\mathbf{A} - \mathbf{A}_1\|_F. \tag{26}$$

Finally, since $\|\mathbf{v}_1\|_1 \geq \|\mathbf{v}_1\|_2 = 1$, the scalar factor $1/(\mathbf{v}_1^\top \mathbf{1}) \leq 1$. Thus:

$$\|\mathbf{A}_1 - \mathbf{A}_{\text{static}}\|_F \leq \sqrt{L}\|\mathbf{A} - \mathbf{A}_1\|_F. \tag{27}$$

**Conclusion:** Substituting back into Eq. 20:

$$
\begin{aligned}
\|\mathbf{Y} - \mathbf{Y}_{\text{static}}\|_F &\leq \|\mathbf{A} - \mathbf{A}_1\|_F \|\mathbf{V}\|_2 + \sqrt{L}\|\mathbf{A} - \mathbf{A}_1\|_F \|\mathbf{V}\|_2 \\
&= (1 + \sqrt{L})\|\mathbf{V}\|_2 \sqrt{\sum_{k=2}^{L} \sigma_k(\mathbf{A})^2}.
\end{aligned}
\tag{28}
$$

$\square$

# B. Implementation Details

## B.1. Dataset Construction Details

Currently, there is no universally accepted benchmark dataset specifically designed for the Image-to-Video (I2V) generation task that provides both high-quality reference images and aligned textual descriptions. Consequently, prior works have relied on self-collected datasets of varying sizes; for instance, Vid-Freeze utilizes a set of 50 images, while I2VGuard employs 300 images.

To ensure a comprehensive and statistically significant evaluation, we constructed a custom benchmark dataset comprising 300 image-text pairs. Our dataset matches the scale of I2VGuard and significantly exceeds that of Vid-Freeze, providing a more robust basis for evaluation. As mentioned in Section 4, the dataset is meticulously balanced across four distinct semantic categories: Humans, Animals, Landscapes, and Objects. Furthermore, it includes a mix of real-world photographs and AI-generated imagery to verify the attack's effectiveness across different domain distributions.

## B.2. Attack Hyperparameters

We summarize the target models and the attack configuration in Table 7. We use a diffusion process with $T$ timesteps and select a single target timestep $t^* = \lfloor 0.8T \rfloor$, which lies in the high-noise regime (closer to $\mathbf{z}_T$ than $\mathbf{z}_0$). For the targeted temporal attention block, we use $l^* = $ Block 5 for 3D U-Net models and $l^* = $ Block 15 for the DiT backbone. We use $K = 100$ PGD iterations for the DiT model and $K = 50$ for the two U-Net models. Regarding conditioning inputs, SVD relies solely on the reference image, whereas DynamiCrafter and LTX-Video utilize identical text prompts for each corresponding image to ensure fair comparison.

Table 7. **Target models and attack hyperparameters.** $T$ denotes the total number of diffusion timesteps of the scheduler.

|  | SVD | DynamiCrafter | LTX-Video |
|---|---|---|---|
| Backbone | 3D U-Net | 3D U-Net | DiT |
| Default frames | 14 | 16 | 25 |
| Default resolution | 576×1024 | 256×256 | 256×256 |
| Target timestep $t^*$ | $\lfloor 0.8T \rfloor$ | $\lfloor 0.8T \rfloor$ | $\lfloor 0.8T \rfloor$ |
| Target block $l^*$ | Block 5 | Block 5 | Block 15 |
| Perturbation budget $\epsilon$ ($\ell_\infty$) | 16/255 | 16/255 | 16/255 |
| Step size $\gamma$ | 0.01 | 0.01 | 0.01 |
| PGD iterations $K$ | 50 | 50 | 100 |

## B.3. Robustness Evaluation Parameters

In Section 4.4, we evaluate the robustness of MEDUSA against three standard input purification defenses. The specific hyperparameters for these transformations are configured as follows:

- **JPEG Compression:** The adversarial images are compressed with a quality factor of 75 (keeping 75% quality).

- **Gaussian Noise:** We apply additive Gaussian noise with a standard deviation of $\sigma = 8/255$ to the adversarial images.

- **Bit-Depth Reduction:** The color depth of the images is quantized from the standard 8 bits to 4 bits per channel.

These are commonly used defense settings in adversarial research. They are strong enough to challenge the perturbation, while keeping the video visually usable.

# C. Additional Optimization Analysis

In Section 3.3, we theoretically argued that directly enforcing a rigid static mapping (Hard-Target) leads to optimization failure due to the vanishing gradient problem inherent in the Softmax mechanism. To empirically validate this, we tracked the $\ell_2$-norm of the gradients with respect to the input perturbation $\delta$ during the optimization process.

We compare two objective functions:

- **Hard-Target:** $\mathcal{L}_{\mathrm{hard}} = \|\mathbf{A} - \mathbf{A}^S\|_F^2$, which forces the attention matrix to match a static template.

- **MEDUSA:** $\mathcal{L}_{\mathrm{nuc}} = \|\mathbf{A}\|_*$, our proposed spectral objective.

Table 8 presents the gradient magnitudes over 50 iterations. The gradients for the Hard-Target objective are remarkably small, staying at the order of $10^{-6}$. This confirms that the optimization starts in the saturated region of the Softmax function, where the Jacobian is vanishingly small, effectively stalling the update of the perturbation $\delta$.

In contrast, MEDUSA maintains robust gradient magnitudes in the range of $10^{-3}$ to $10^{-2}$. The nuclear norm acts on the singular values directly, providing a consistent gradient pressure that does not suffer from element-wise saturation. This robust gradient signal allows the PGD optimizer to effectively navigate the optimization and induce the desired temporal rank collapse.

*Table 8.* **Comparison of Gradient Magnitudes.** We report the $\ell_2$-norm of the gradient with respect to the perturbation $\delta$ at different PGD iterations. The Hard-Target objective suffers from effective gradient vanishing ($\sim 10^{-6}$), whereas MEDUSA maintains significant gradient magnitudes, enabling effective optimization.

| Iteration $k$ | $\|\nabla_\delta \mathcal{L}_{\mathrm{hard}}\|_2$ (Hard-Target) | $\|\nabla_\delta \mathcal{L}_{\mathrm{nuc}}\|_2$ (MEDUSA) |
|---|---|---|
| 1 | $8.93 \times 10^{-6}$ | $2.90 \times 10^{-3}$ |
| 10 | $6.79 \times 10^{-6}$ | $7.02 \times 10^{-3}$ |
| 20 | $7.26 \times 10^{-6}$ | $6.36 \times 10^{-3}$ |
| 30 | $4.67 \times 10^{-6}$ | $1.01 \times 10^{-2}$ |
| 40 | $6.27 \times 10^{-6}$ | $3.74 \times 10^{-3}$ |
| 50 | $6.49 \times 10^{-6}$ | $8.44 \times 10^{-3}$ |

## D. Additional Qualitative Results

In this section, we provide additional visualizations to further evaluate the performance of MEDUSA. First, we assess the visual imperceptibility of our attack by comparing it with random noise (Figure 5). Second, we provide an additional comparative example against baseline methods to reinforce our method (Figure 6). Finally, we present a visual comparison between our method and the clean generation across diverse scene categories (Figure 7).

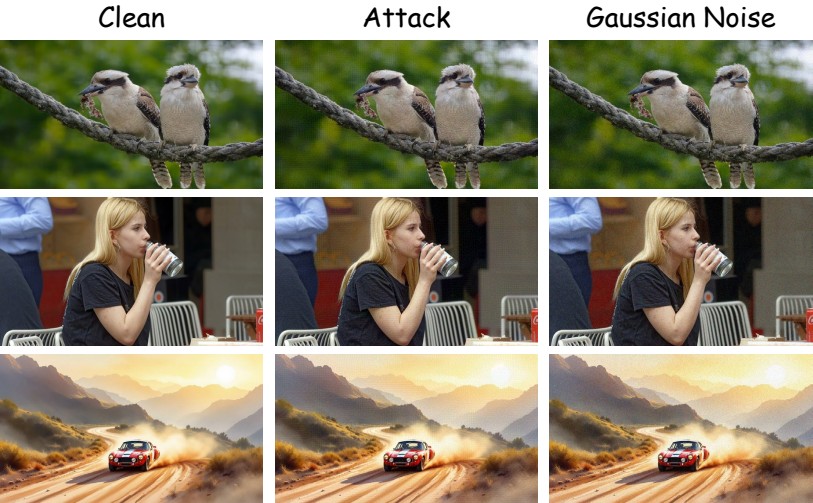

*Figure 5.* **Visual imperceptibility comparison.** From left to right: the clean reference image, the adversarial image generated by MEDUSA ($\epsilon = 16/255$), and the image with random Gaussian noise ($\epsilon = 8/255$). Our spectral perturbation remains highly imperceptible to the human eye, exhibiting fewer visible artifacts than random Gaussian noise, even though our method utilizes a larger perturbation budget.

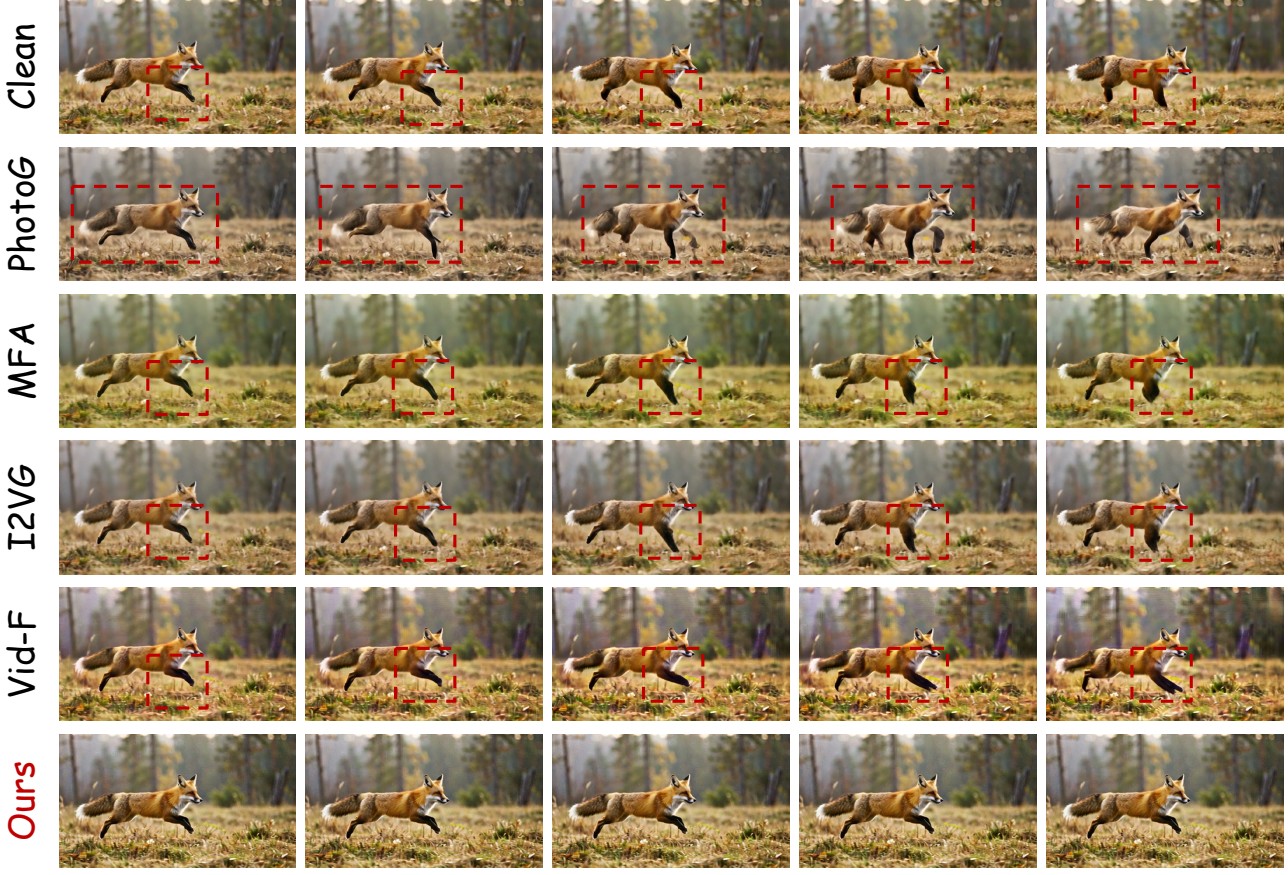

*Figure 6.* **Additional qualitative comparison with baselines.** We compare MEDUSA against PhotoGuard, MFA, I2VGuard, and Vid-Freeze on a dynamic scene. Consistent with the results in the main text, existing attacks fail to suppress motion. In contrast, MEDUSA effectively eliminates temporal dynamics while preserving high visual fidelity.

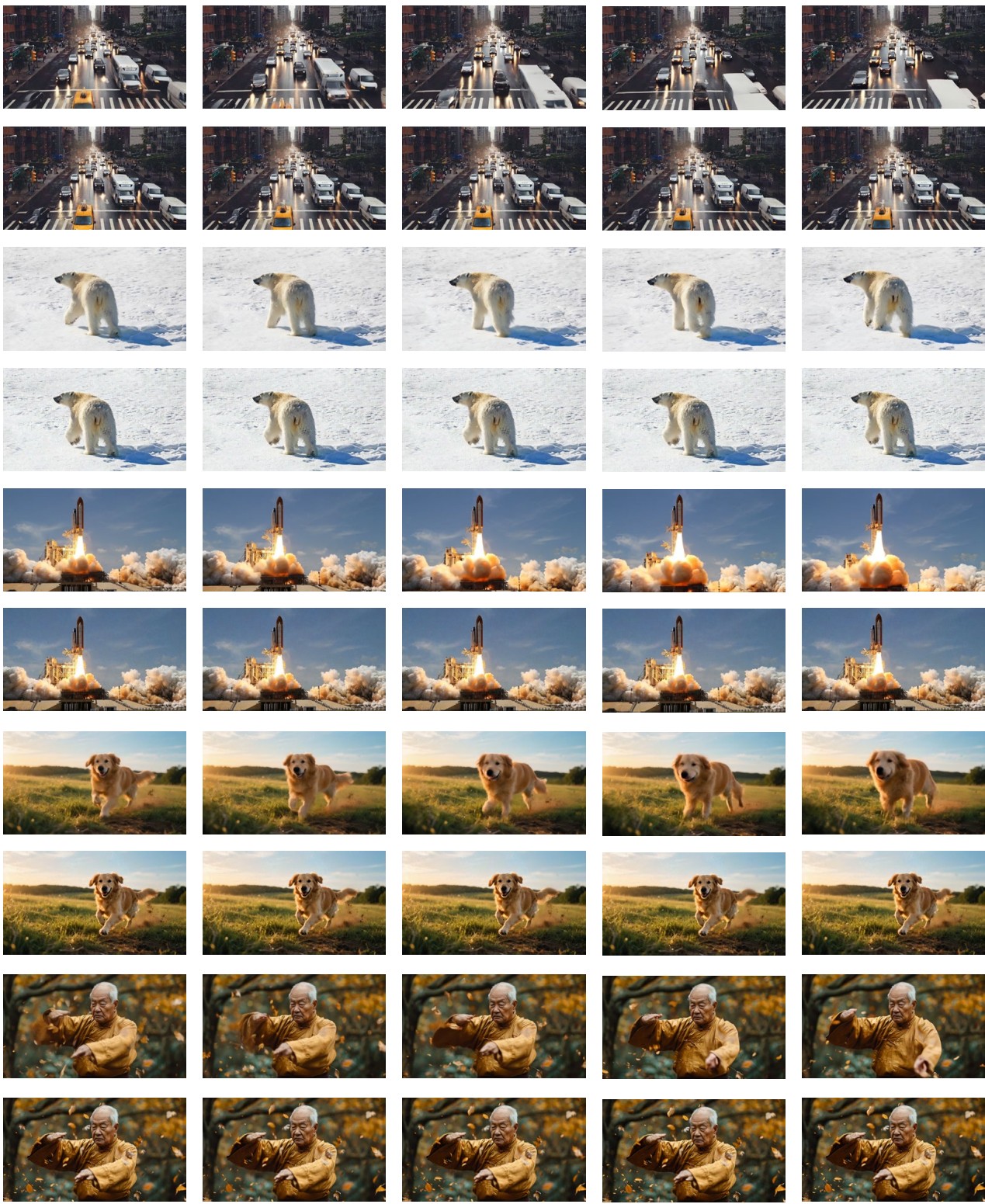

*Figure 7.* **Gallery of MEDUSA results across diverse categories.** For each example, the *top row* shows the video generated from the clean reference image, while the *bottom row* shows the video generated from the attacked reference image produced by MEDUSA. Across landscapes, animals, and human portraits, MEDUSA consistently suppresses temporal dynamics and produces near-static outputs.

