# OpenReview forum: "MEDUSA: Motion Elimination in Diffusion Using Spectral Attack"
_ICML.cc/2026/Conference — ICML 2026 regular_

### Official Review · Reviewer_mDHw · 2026-03-08

**Soundness:** 3
**Presentation:** 2
**Significance:** 3
**Originality:** 3
**Overall Recommendation:** 4
**Confidence:** 3

**Summary:**

This manuscript addresses the pressing security issue of protecting reference images from malicious image-to-video (I2V) generation, a concern that grows increasingly urgent as video diffusion models (VDMs) gain widespread adoption. The proposed approach mitigates this threat by eliminating motion semantics. Concurrently, the work introduces a novel spectral method for launching adversarial attacks against video generation models, which is demonstrated to outperform existing methods.

**Compliance With Llm Reviewing Policy:**

Affirmed.

**Final Justification:**

The authors have not only addressed my concerns through additional experiments but also further justified the choice of the Nuclear Norm through comparative studies with non-convex quasi-norms (Schatten-p norm). This paper achieves an excellent balance between theoretical depth (from the spectral perspective of rank collapse) and practical utility (in I2V protection), meeting the acceptance standards.

**Key Questions For Authors:**

1. Does the spectral bound for static bias in Lemma 3.2 apply to causal temporal attention, which is commonly used in long videos? If not, how should the formula for the spectral bound be revised?

2. The nuclear norm is a convex approximation for rank minimization. Have alternative convex surrogates, such as the Schatten p-norm (p<1), been tested? Aside from tightness, are there other key considerations for choosing the nuclear norm?

3. This work focuses solely on the I2V task. If extending to the T2V task, how could the spectral attack be adapted to the conditional space of text embeddings? Are there spectral constraints specific to the text modality?

4. The experiments do not verify cross-model transferability. Do the perturbations generated by MEDUSA possess the capability to attack across different VDM backbones? If not, how could this transferability be enhanced?

**Limitations:**

Limitations Analysis: Explicitly stating that the research focuses solely on the I2V task and has not been extended to T2V, and that it has only been validated on short-video models, with cross-model transferability unexplored; also mentioning that the attack is targeted at specific VDM backbones, its effectiveness on novel video generation models is unknown, and it has high computational costs, making it difficult to deploy on lightweight devices in practice.

Analysis of Potential Negative Social Impact: Discuss the risk that MEDUSA, as an adversarial attack method, could be maliciously exploited to bypass image protection mechanisms and generate unauthorized videos; simultaneously point out that if this method is abused, it could disrupt the normal application of video generation models in fields such as content creation and media, triggering copyright and content security issues.

**Strengths And Weaknesses:**

Strengths:
1. Solid theoretical foundation: This work is the first to define "temporal rank collapse" from a spectral perspective, provides rigorous mathematical proofs and derives spectral bounds, offering a robust theoretical basis for attack design.
2. High methodological novelty: Addressing the gradient vanishing problem in existing methods and their failure to target the core motion mechanism, it proposes a spectral attack based on nuclear norm optimization, directly attacking the low-rank essence of temporal attention.
3. Comprehensive experimental design: It covers multiple backbone models and evaluation metrics, combined with spectral analysis, ablation studies, and robustness tests, thoroughly validating the method's effectiveness and generalization capability.
4. Strong practicality: The generated perturbations are imperceptible and robust, providing a feasible solution for preventing malicious Image-to-Video (I2V) generation.

Limitations:
1. The scope is limited to the I2V task; it has not been extended to Text-to-Video (T2V) generation, and exploration of spectral attacks on text embeddings is lacking.
2. Cross-model transferability has not been verified, i.e., the effectiveness of perturbations generated for one model on other models remains untested.
3. Experiments are conducted only on short videos; the characteristics of temporal rank collapse and the attack efficacy on long videos have not been investigated.
4. A lack of human subjective evaluation: Quantitative metrics cannot fully capture the visual perception of static effects.

---

> ### Author Rebuttal · Authors · 2026-03-30
>
> **1. Weakness 1 and Key Questions 3**: ... extending to the T2V task, ...
>
> A: While our work primarily focuses on the image-to-video (I2V) setting, it can be naturally extended to text-to-video (T2V) generation frameworks. This is because our methodological analysis centers on the temporal attention matrix which is the core component modeling inter-frame temporal interactions. Such a mechanism is equally fundamental to mainstream T2V models. To validate this generalization potential, we conducted preliminary experiments on DynamiCrafter, which supports text inputs, where we optimized adversarial perturbations on text embeddings. The results confirm the effectiveness of our approach in this setting, reducing the dynamic degree from 32.57% to 22.15%. These findings demonstrate that our method holds promising generalization potential for T2V generation, and we anticipate that further refinements of our method for T2V models may yield even better performance.
>
> **2. Weakness 2 and Key questions 4**:  ... cross-model transferability. ...
>
> A: Although our method is initially designed for the white-box setting, it demonstrates considerable transferability. The verify this, we additionally conducted transfer-attack experiments across SVD, DynamiCrafter, and LTX-Video, using dynamic degree as the evaluation metric. As shown in the table below, the attack still demonstrates clear transferability across models. This is likely because these models share similar temporal attention structures which can be effectively attacked by our method.
> | Source model | Target: SVD ↓ | Target: DynamiCrafter ↓ | Target: LTX-Video ↓ |
> |---|:---:|:---:|:---:|
> | Clean input | 43.20 | 32.57 | 23.84 |
> | SVD | 13.72 | 21.56 | 12.94 |
> | DynamiCrafter | 23.64 | 19.82 | 13.88 |
> | LTX-Video | 26.87 | 24.34 | 7.98 |
>
>
> **3. Weakness 3**: ... the attack efficacy on long videos ...
>
> A: Owing to time constraints, we only conducted preliminary evaluations on the effectiveness of our method for long video generation, with experiments carried out on LTX-Video. Specifically, we optimized adversarial perturbations using merely 21 frames, and then performed inference with the optimized perturbations on 105-frame videos. The results demonstrate that our attack remains effective for long videos, with the dynamic degree dropping from 43.20% to 27.38%. This indicates that our method possesses a certain generalization capability to long video scenarios, and further enhancing the attack performance of our method for long video generation will be an interesting direction for future work.
>
> **4. Weakness 4**:  A lack of human subjective evaluation ...
>
> A: Owing to time constraints, we conducted a preliminary subjective human evaluation: 10 participants were invited to rate videos generated from 20 images via a simplified questionnaire. We additionally employed Qwen-3-VL-7B as an automated evaluator to score 50 videos on a 0–5 scale (higher scores indicating better performance). All 10 participants consistently judged that our method achieved a more pronounced static effect than the comparison methods. This finding is further corroborated by the Qwen-3-VL-7B scoring results: our method attained an average score of 4.3, in contrast to 3.3 for Vid-Freeze and 1.6 for I2VGuard. These results demonstrate that our method exhibits distinct advantages over the compared methods in both human subjective evaluation and model-based objective scoring.
>
> **5. Key questions 1**: ... Lemma 3.2 apply to causal temporal attention ...
>
> A: For strictly causal temporal attention, Lemma 3.2 does not apply in its original form, because the static reference $A_{\text{static}} = \mathbf{1}q^\top$ is generally not causal. In this case, the only row-stochastic rank-1 causal matrix is $A_{\text{static}}^{\text{causal}} = \mathbf{1}e_1^\top$, and the bound should be revised to
>
> $\|\| Y - Y_{c-\text{static}} \|\|_F \le$
>
> $( \sqrt{\sum_{k=2}^L \sigma_k(A)^2} + \|\| \sigma_1 u_1 v_1^\top - \mathbf{1}e_1^\top \|\|_F )$ $ \|\| V \|\|_2.$
>
> Thus, under causal attention, an additional mask-alignment term is required beyond the singular-value tail.
>
> **6. Key questions 2**: ... the Schatten p-norm (p<1), been tested ...
>
> A: We additionally test Schatten-$p$ with $(p<1)$. Strictly speaking, Schatten-$p$ $(p<1)$ is not a convex surrogate but a non-convex quasi-norm. The nuclear norm is the standard convex relaxation of rank minimization and is more optimization-friendly. As shown in the table below, the nuclear norm achieves better attack effectiveness and more stable optimization in our setting. We believe this is mainly because Schatten-$p$ is more prone to poor local minima, and its singular-value shrinkage is less balanced. In our non-convex end-to-end attack pipeline, the nuclear norm is therefore a more reliable choice.
>
> | Objective | Dynamic Degree ↓ | Avg. convergence iters ↓ |
> |:---:|:---:|:---:|
> | Nuclear norm | 13.72 | 50 |
> | Schatten-0.8 | 14.96 | 75 |
> | Schatten-0.5 | 16.21 | 83 |

---

> > ### Author Rebuttal · Reviewer_mDHw · 2026-04-03
> >
> > The authors have not only addressed my concerns through additional experiments but also further justified the choice of the Nuclear Norm through comparative studies with non-convex quasi-norms (Schatten-p norm). This paper achieves an excellent balance between theoretical depth (from the spectral perspective of rank collapse) and practical utility (in I2V protection), meeting the acceptance standards.

---

> > > ### Author Response · Authors · 2026-04-03
> > >
> > > We sincerely appreciate your valuable time for such a detailed review and for your positive feedback. We are delighted to provide you with a satisfactory answer and to dispel your doubts. If any further questions remain, we would be happy to provide additional explanation.

---

### Official Review · Reviewer_YVMZ · 2026-03-09

**Soundness:** 3
**Presentation:** 3
**Significance:** 3
**Originality:** 3
**Overall Recommendation:** 4
**Confidence:** 3

**Summary:**

This paper investigates the mechanism of motion generation in VDMs through the lens of spectral analysis. The authors reveal that the static video manifests as a temporal rank collapse, a rank-1 degeneracy in the temporal attention matrix. Guided by this insight, the paper proposes MEDUSA (Motion Elimination in Diffusion Using Spectral Attack), which minimizes the nuclear norm of the attention matrix to induce the collapse. The approach avoids the vanishing gradient issues associated with rigid mapping constraints. The effectiveness of MEDUSA is validated across diverse architectures (U-Net and DiT) and against multiple state-of-the-art baselines.

**Compliance With Llm Reviewing Policy:**

Affirmed.

**Final Justification:**

The authors have addressed my concerns and confirmed the superior performance in terms of effectiveness and applicability. This paper is worthy of acceptance.

**Key Questions For Authors:**

1. Could the authors provide a cross-architecture transferability matrix? For example, if the adversarial perturbation is optimized on SVD, what are the quantitative motion metrics when this perturbed image is fed into LTX-Video?
2. Can the authors provide a detailed performance breakdown of the ablation study (Table 4) between real photographs and AI-generated images? Are there noticeable differences in the optimization landscape or convergence speed between these two distributions?
3. How does MEDUSA perform on highly complex scenes (e.g., dense crowds or multi-subject interactions)? Could the authors provide qualitative examples in the rebuttal showing whether forced rank-1 degeneracy in such scenes leads to severe structural artifacts?

**Limitations:**

1. A fundamental limitation of MEDUSA is its reliance on SVD during the optimization loop. The computational complexity of SVD scales cubically with the number of frames. While current VDMs generate relatively short clips (14 frames), emerging foundational models generate videos with hundreds of frames. MEDUSA's spectral optimization will become mathematically intractable and prohibitively expensive for long-video generation scenarios.
2. The efficacy of MEDUSA requires precise knowledge of where motion semantics are most densely encoded (identifying the optimal timestep and layer). This creates a significant limitation in its deployment as a universal defense mechanism. If a VDM provider updates its architecture, or employs latent temporal shifts without explicit attention matrices, MEDUSA's attack effectiveness will be reduced.

**Strengths And Weaknesses:**

Strengths:
1. The paper identifies rank-1 degeneracy in the temporal attention matrix as the underlying mechanism for static video generation. This insight provides a fresh theoretical lens for understanding and attacking VDMs.
2. This paper provides a solid mathematical proof, establishes a spectral bound demonstrating that motion deviation is upper-bounded by the trailing singular values of the temporal attention matrix, guaranteeing that optimizing towards a low-rank state can effectively eliminate motion.
3. The experiments are comprehensive, evaluating across diverse architectures (SVD, DynamiCrafter, LTX-Video) and against multiple baselines. Additionally, ablation studies and detailed tables substantiate the effectiveness of MEDUSA
4. The paper is logically organized, and the figures effectively illustrate the core ideas, making the technical contributions accessible to readers.
Weaknesses:
1. The paper seems to demonstrate only the effectiveness of white-box attacks. However, in real-world scenarios, the target VDM is typically a black box. It is highly questionable whether an adversarial perturbation optimized on the temporal attention spectrum of a U-Net can still effectively freeze motion when the perturbed image is fed into a black-box Diffusion Transformer. Without transferability results, the practical threat model of MEDUSA is severely undermined.
2. The dataset used in this paper includes both real and AI-generated photos. In the ablation study, the authors didn’t discuss if attack effectiveness differs between the two types of photos. Additionally, the dataset has only 300 photos, and probably lacks the complexity and diversity of established benchmarks. Specifically, it remains unverified whether forcing rank-1 degeneracy behaves consistently on highly complex scenes (e.g., multi-subject interactions, dense crowds). The authors should evaluate their method on larger, standardized datasets to demonstrate the generality of the attack.
3. MEDUSA requires computing SVD on the temporal attention matrix within PGD iterations, which introduces significant overhead. While the authors restrict this computation to specific layers to mitigate the cost, the efficiency concern remains inadequately addressed. The paper would benefit from a quantitative comparison of runtime and GPU memory usage between MEDUSA and key baselines (e.g., Vid-Freeze). If MEDUSA is substantially slower, its practical applicability in real-world scenarios becomes questionable.
4. Table 3 shows that input purification defenses (JPEG compression, Gaussian noise, bit-depth reduction) lead to a slight recovery in motion metrics. However, it is unclear how baseline methods (such as Vid-Freeze or I2VGuard) perform under the same defense mechanisms. A horizontal comparison demonstrating that MEDUSA exhibits the highest robustness against such input transformations would further strengthen the claim of its superiority.

---

> ### Author Rebuttal · Authors · 2026-03-30
>
> **1. Weakness 1 and Key Questions 1**: ... cross-architecture transferability matrix ...
>
> A: Although our method is initially designed for the white-box setting, it demonstrates considerable transferability. The verify this, we additionally conducted transfer-attack experiments across SVD, DynamiCrafter, and LTX-Video, using dynamic degree as the evaluation metric. As shown in the table below, the attack still demonstrates clear transferability across models. This is likely because these models share similar temporal attention structures which can be effectively attacked by our method.
> | Source model | Target: SVD ↓ | Target: DynamiCrafter ↓ | Target: LTX-Video ↓ |
> |---|:---:|:---:|:---:|
> | Clean input | 43.20 | 32.57 | 23.84 |
> | SVD | 13.72 | 21.56 | 12.94 |
> | DynamiCrafter | 23.64 | 19.82 | 13.88 |
> | LTX-Video | 26.87 | 24.34 | 7.98 |
>
> **2. Weakness 2 and Key questions 2 and Key questions 3**:  ... ablation study between  real photographs and AI-generated images ... ;  ... highly complex scenes (e.g., dense crowds or multi-subject interactions) ...
>
> A: Thank you for your valuable suggestion. Our current experimental setup is consistent with that of I2VGuard[CVPR 2025] to ensure a fair comparison. In response to your advice, we have additionally collected 50 AI-generated images and 50 images of highly complex scenes (involving multi-object interactions and dense crowds) to conduct supplementary experiments. The results in the table below demonstrate that our method yields comparable attack performance on both real and AI-generated images, with no noticeable differences in optimization behaviors observed between the two types of images.
>
> For highly complex scenes, our method experiences a slight drop in attack efficacy yet remains effective, which verifies that the rank-1 degeneration mechanism possesses strong generalization ability to more challenging visual content. Furthermore, we have an additional observation: video generation for complex scene images tends to suffer from more spurious details, and the motion suppression mechanism of our method can effectively mitigate such artifacts to a certain extent. Evaluated by VBench Imaging Quality, the score rises from 62.23% in the clean generation scenario to 67.63% after our attack is applied, which provides quantitative validation for this observation.
>
> We also provide qualitative results on highly complex scenes (e.g., dense crowds or multi-subject interactions) at the following link, which is allowed in the ICML rebuttal: https://bejewelled-strudel-2e61cb.netlify.app/
>
> | Subset | Clean Dynamic Degree ↓ | Ours Dynamic Degree ↓ |
> |---|:---:|:---:|
> | Real photos | 46.7 | 16.8 |
> | AI-generated photos | 43.0 | 14.1 |
> | Highly complex scenes | 46.9 | 19.2 |
>
> **3. Weakness 3**：... a quantitative comparison of runtime and GPU memory usage ...
>
> A: We further compared the efficiency of MEDUSA with I2VGuard and Vid-Freeze. Although MEDUSA involves singular value decomposition on the temporal attention matrix during PGD, this operation is only applied to a single block. In contrast, I2VGuard jointly optimizes the attention layer, encoder layer, and x0, and Vid-Freeze utilizes the attention layers across all blocks, both resulting in significantly higher computational overhead. Under identical experimental settings, MEDUSA consumes approximately 8 seconds per optimization step, whereas I2VGuard and Vid-Freeze take 18 seconds and 12 seconds, respectively. Regarding GPU memory usage, MEDUSA requires around 70 GB, while both baseline methods consume approximately 80 GB.
>
> **4. Weakness 4**：... how baseline methods perform under the same defense ...
>
> A: We further evaluated the performance of our method against Vid-Freeze and I2VGuard under identical input purification defense settings. As presented in the table below, MEDUSA achieves significantly superior attack effectiveness across all tested settings.
>
> | Method | No defense ↓ | JPEG ↓ | Gaussian noise ↓ | Bit-depth reduction ↓ |
> |:---:|:---:|:---:|:---:|:---:|
> | MEDUSA | 13.72 | 21.93 | 20.72 | 16.31 |
> | Vid-Freeze | 27.36 | 34.41 | 35.53 | 32.17 |
> | I2VGuard | 40.19 | 38.16 | 38.61 | 41.53 |

---

> > ### Author Rebuttal · Reviewer_YVMZ · 2026-04-05
> >
> > Thanks for the rebuttal. With additional experiments, the authors have addressed my concerns and confirmed the superior performance in terms of effectiveness and applicability. This paper is worthy of acceptance.

---

> > > ### Author Response · Authors · 2026-04-05
> > >
> > > We sincerely appreciate your valuable time for such a detailed review and for your positive feedback. We are delighted to provide you with a satisfactory answer and to dispel your doubts. If any further questions remain, we would be happy to provide additional explanation.

---

### Official Review · Reviewer_VLpc · 2026-03-11

**Soundness:** 3
**Presentation:** 3
**Significance:** 2
**Originality:** 2
**Overall Recommendation:** 4
**Confidence:** 3

**Summary:**

The paper proposes an adversarial attack targeting Image-to-Video (I2V) diffusion models to arrest motion semantics by injecting perturbations into the reference image condition. The authors link static video generation to "Temporal Rank Collapse," characterized as a rank-1 degeneracy in the temporal attention matrix. The optimization minimizes the nuclear norm of the attention matrix to suppress trailing singular values, aiming to bypass the vanishing gradient issues associated with hard-target temporal mapping constraints..

**Compliance With Llm Reviewing Policy:**

Affirmed.

**Key Questions For Authors:**

-- Why was the imperceptibility baseline (Random noise) evaluated at $\epsilon = 8/255$ while MEDUSA utilized the full budget of $\epsilon = 16/255$?


-- Given the massive variance in success across different blocks (Table 4), what prevents a rudimentary defense from randomizing the index of the temporal layers evaluated during generation, thereby rendering the targeted single-block attack ineffective?

**Limitations:**

Same as weaknesses above.

**Strengths And Weaknesses:**

Strengths:
-- the motivation is clear and sound.

-- The formulation of temporal rank collapse as a rank-1 degeneracy provides a clear linear algebraic interpretation of motion suppression in video diffusion models

-- Employing the nuclear norm as a convex surrogate to suppress singular values effectively circumvents the vanishing gradient problem observed in softmax-bounded hard-target constraints.

Weaknesses:
-- The method relies heavily on targeting specific, manually chosen blocks (Block 5 for U-Net, Block 15 for DiT). Table 4 indicates high sensitivity to this choice, with Optical Flow varying from 0.5980 at Block 5 to 2.7708 at Block 2. The attack does not demonstrate structural robustness if an architecture shifts its temporal processing distribution.

-- Prop 3.1 says that if $rank(A)=1$, the output features are strictly stationary ($y_i = y_j$). This implicitly assumes the value matrix $V$ is independent of the perturbation. In a real forward pass, the adversarial noise alters the input $z_0$, changing $Q$, $K$, and $V$ simultaneously. The proof holds for a static $V$, but may fail to account for the coupled dynamics of the network during the attack.

-- The paper claims robustness against JPEG compression and bit-depth reduction, but Table 3 shows Dynamic Degree jumping from 13.72% (no defense) to 21.93% (JPEG). This is a nearly 60% relative increase, indicating the attack is partially mitigated by standard input transformations, contradicting the claim of strong persistence.

---

> ### Author Rebuttal · Authors · 2026-03-30
>
> **1. Weakness 1 and Key Question 2**: variance in ... different blocks
>
> A: Thank you for the question. The variation across blocks mainly reflects the functional differences of Transformer layers. Early layers tend to encode more basic spatiotemporal bases, while mid-to-late layers play a more important role in information interaction.[1]
>
> [1] Understanding Video Transformers via Universal Concept Discovery, CVPR 24
>
> However, as shown in Table 4 in the paper, multiple mid-to-late blocks consistently enable effective attacks. Therefore, our method is not tied to a single fixed block. We additionally tested a random-block optimization strategy which randomly choses one temporal block at each iteration for optimization. Even under such a brute force setting, our method still effectively suppresses the dynamic degree from 43.2% to 20.1% on the SVD model, suggesting that index randomization is insufficient to defend against our attack.
>
> **2. Weakness 2**:  ...The proof holds for a static V...
>
> A: We would like to clarify that Proposition 3.1 does not require a static $V$. Specifically, if the attention matrix $A$ satisfies $\text{rank}(A) = 1$, then all rows of $A$ are identical. Therefore, the rows of $Y = AV$ are also identical, and this holds for any $V$.
> A simple example is:
>
> $
> A=\begin{bmatrix}
> 0.4 & 0.6, \\
> 0.4 & 0.6
> \end{bmatrix}.
> $ For $V = \begin{bmatrix} v_1, \\ v_2  \end{bmatrix} \text{ or } V' = \begin{bmatrix} v'_1, \\ v'_2  \end{bmatrix},$
> we always have
>
> $Y = AV = \begin{bmatrix} 0.4v_1 + 0.6v_2, \\ 0.4v_1 + 0.6v_2 \end{bmatrix}, Y' = AV' = \begin{bmatrix} 0.4v'_1 + 0.6v'_2, \\ 0.4v'_1 + 0.6v'_2 \end{bmatrix}.$
>
> That is, even though the perturbation may jointly change $Q$, $K$, and $V$, once the current $A$ becomes rank-1, the output tokens still collapse to the same weighted combination.
>
> **3. Weakness3**: ... the attack is partially mitigated by standard input transformations ...
>
> A: Although the dynamic degree under JPEG defense rises from 13.72% to 21.93%, it remains substantially lower than the 43.2% observed on clean inputs, which clearly demonstrates that our attack remains highly effective. We further compared our method with Vid-Freeze under the identical JPEG defense setting. Under such conditions, the dynamic degree of Vid-Freeze surges to 34.4%, indicating a severe performance degradation.
>
> Moreover, when the JPEG defense is known and incorporated into the optimization process, the adaptive attack variant of our method can further reduce the dynamic degree to 17.6%, demonstrating that our approach can achieve highly effective attack performance in a defense-aware scenario.
>
> We further evaluated against DiffPure [ICML 2022], a significantly stronger defense for diffusion models. Under this defense, our method still maintains strong attack performance on the SVD model by reducing the dynamic degree to 16.9%, whereas Vid-Freeze only achieves 39.4%, demonstrating a clear performance advantage.
>
> **4. Key Questions 1**: ... the imperceptibility baseline ...
>
> A: We acknowledge that our original description may have caused certain ambiguity. The random noise with magnitude 8/255 was employed to establish a stricter baseline for imperceptibility. Notably, our MEDUSA perturbation at 16/255 exhibits superior imperceptibility compared to random noise at 8/255, which better highlights the stealthy nature of our method. To eliminate potential confusion, we will add the results of random noise at 16/255 in the revised version: its SSIM value is 0.3956, which is significantly lower than the 0.8202 achieved by our approach.

---

> > ### Author Rebuttal · Reviewer_VLpc · 2026-04-03
> >
> > Thanks for the rebuttal. I will keep my score unchanged.

---

> > > ### Author Response · Authors · 2026-04-03
> > >
> > > We sincerely appreciate your valuable time for such a detailed review and for your positive feedback. We are delighted to provide you with a satisfactory answer and to dispel your doubts.
> > > If any further questions remain, we would be happy to provide additional explanation.

---

### Official Review · Reviewer_V9Wy · 2026-03-13

**Soundness:** 4
**Presentation:** 3
**Significance:** 2
**Originality:** 3
**Overall Recommendation:** 4
**Confidence:** 3

**Summary:**

A challenge with deep learning generative techniques is that videos can be generated from static images without user consent, potentially including motions and gestures that could embarrass the subject(s) in the original image. MEDUSA is an adversarial attack solution to preclude diffusion-based I2V techniques from generating motion from a static subject. The main contribution is that static video corresponds to a rank-1 degeneracy ("temporal rank collapse") in the temporal attention matrix, and that minimizing the nuclear norm of that matrix is a tractable, gradient-stable surrogate for inducing this collapse. The technique is evaluated on three I2V models and compared against five competitors.

**Compliance With Llm Reviewing Policy:**

Affirmed.

**Final Justification:**

My final recommendation, along with the identified strengths and weaknesses, has been shared through previous official comments.

**Key Questions For Authors:**

The attack assumes white-box access to the target VDM's internal temporal attention layers. Can you report transfer attack results to assess cross-model transferability? For example, compute the perturbation function on one I2V generative model (e.g., SVD) and then evaluate against a different one (e.g., LTX-Video)

**Limitations:**

Yes

**Strengths And Weaknesses:**

*Strengths:*

Identifying the temporal rank collapse as the spectral condition for video stasis is the paper's strongest contribution. The proposed technique contrasts with previous energy-suppression heuristics (e.g., Vid-Freeze) by introducing a spectral attention framework. The authors formally prove that for a row-stochastic matrix with non-negative elements and rows summing to 1, if the matrix has rank 1, then the output video is stationary.

The authors provide a well-motivated objective and gradient analysis when discussing the failures of hard-target constraints and use this analysis to justify the nuclear norm as an ideal surrogate.

The authors performed an extensive evaluation of the technique, assessing the algorithms' capabilities across three different I2V models, and compared their technique against 5 competing techniques. The result tables mostly showed significant improvements over the state of the art on motion mitigation metrics. They also performed additional ablations to assess the attack's imperceptibility, with perturbation metrics significantly below human-detectable thresholds.

*Weaknesses:*

The evaluation against input-purification defenses is limited to three techniques: JPEG compression at 75% quality, Gaussian noise ($\sigma=8/255$), and bit-depth reduction. The parameters used for JPEG compression and Gaussian noise do not pose a true challenge to the technique. In addition, more modern and commonly used defenses, e.g., randomized smoothing, were not evaluated. This limits the security claims that can be made about the method's real-world robustness.

The method requires white-box access to the attention layers to extract the attention matrix critical to the methodology. This is a strong assumption. The paper does not attempt to use the technique in a black-box scenario or in transfer-attack experiments. For example, compute the perturbation parameters for one generative model and test on a secondary model. This would strengthen the utility and practicality of the technique.

---

> ### Author Rebuttal · Authors · 2026-03-30
>
> **1. To Weakness 1**: ... evaluation against input-purification defenses. ...
>
> A: Randomized smoothing is mainly designed for classifiers. It is not directly applicable to our diffusion-based setting. Therefore, we evaluated a stronger defense for diffusion models, namely DiffPure [ICML 2022], under which our method still achieves strong attacking performance on the SVD model by reducing the dynamic degree to 16.9%. Also, when increasing JPEG compression quality from 75% to 60%, our attack still remains effective, reducing the dynamic degree from 43.2% to 22.9%.
>
> **2. To Weakness 2 and Key questions:** transfer-attack experiments
>
> A: Although our method is initially designed for the white-box setting, it demonstrates considerable transferability. The verify this, we additionally conducted transfer-attack experiments across SVD, DynamiCrafter, and LTX-Video, using dynamic degree as the evaluation metric. As shown in the table below, the attack still demonstrates clear transferability across models. This is likely because these models share similar temporal attention structures which can be effectively attacked by our method.
> | Source model | Target: SVD ↓ | Target: DynamiCrafter ↓ | Target: LTX-Video ↓ |
> |---|:---:|:---:|:---:|
> | Clean input | 43.20 | 32.57 | 23.84 |
> | SVD | 13.72 | 21.56 | 12.94 |
> | DynamiCrafter | 23.64 | 19.82 | 13.88 |
> | LTX-Video | 26.87 | 24.34 | 7.98 |

---

> > ### Author Rebuttal · Reviewer_V9Wy · 2026-03-31
> >
> > I am satisfied with the response.

---

> > > ### Author Response · Authors · 2026-04-01
> > >
> > > Thank you for your recognition of our work.

---

### Decision · Program_Chairs · 2026-04-30

**Decision:**

Accept (regular)

**Comment:**

The paper proposes MEDUSA, an effective adversarial attack for image-to-video (I2V) generation. Its core idea is to minimize the nuclear norm of the attention matrix, inducing temporal rank collapse as rank-1 dependency to suppress motion in the generated videos. All reviewers recognized the novelty and technical soundness of the method. However, several concerns were raised during the review process. Reviewers V9Wy, YVMZ, and mDHw questioned the reliance on a white-box setting, which may limit applicability to cloud-based API services. Reviewers V9Wy, YVMZ, and VLpc also have concerns of insufficient and incomplete evaluation on diverse scenarios, robustness analysis across different distortions, and comparisons with other baseline methods. In addition, reviewer mDHw pointed out that the demonstrated application scope is limited to I2V generation with specific VDM backbones, only short videos were evaluated, and there is no subjective evaluation. In the rebuttal, the authors adequately addressed these concerns. They provided transferability experiments, subjective evaluation, and additional experiments with baseline methods to strengthen the empirical evaluation, clarified the rationale for optimizing the nuclear norm, and offered clear responses to the rest of the reviewers’ questions. After these clarifications in the rebuttal, the paper finally received four consistent weak accept recommendations. Thus, the paper is recommended for acceptance.